# Invasive acupuncture for gastroparesis after thoracic or abdominal surgery: a systematic review and meta-analysis

Zhenjia Fan ,[1] Yuqin Qiu ,[2] Xuewei Qi,[1] Jingnan Xu,[1] Yuxiang Wan,[1] Yingxu Hao,[3] Wenquan Niu,[4] Jinchang Huang[1]

[1]Department of minimally invasive acupuncture Oncology, Third Affiliated Hospital, Beijing University of Chinese Medicine, Beijing, China
[2]Department of Integrated Traditional Chinese and Western Medicine, The First Affiliated Hospital of Chongqing Medical University, Chongqing, China
[3]Oncology Department of Integrated Traditional Chinese and Western Medicine, China-Japan Friendship Hospital, Beijing, China
[4]Center for Evidence-Based Medicine, Capital Institute of Pediatrics, Beijing, China

**Correspondence to**
Dr Jinchang Huang;
zryhhuang@163.com

## ABSTRACT

**Objectives** This meta-analysis aimed to systematically evaluate the efficacy of acupuncture in treating postsurgical gastroparesis syndrome (PGS) after thoracic or abdominal surgery.

**Design** Systematic review and meta-analysis.

**Data sources** Twelve databases (PubMed, Embase, Cochrane Library Cochrane Central Register of Controlled Trials (CENTRAL), Medline (Ovid) (from 1946), Web of Science, EBSCO, Scopus, Open Grey, China National Knowledge Infrastructure (CNKI), Wanfang Database, Chinese Scientific Journals Database (VIP) and China Biology Medicine disc (CBM)) and three registration websites (WHO International Clinical Trials Registry Platform (ICTRP), ClinicalTrials.gov, and Chinese Clinical Trial Registry (ChiCTR)) were searched from the inception to September 2022, and citations of the included literature were screened.

**Eligibility criteria** All randomised controlled trials addressing invasive acupuncture for PGS.

**Data extraction and synthesis** Key information on the included studies was extracted by two reviewers independently. Risk ratio (RR) with 95% CI was used for categorical data, and mean difference with 95% CI for continuous data. The quality of evidence was assessed using Grading of Recommendations Assessment, Development and Evaluation. Outcomes were conducted with trial sequential analysis (TSA).

**Results** Fifteen studies with 759 patients met the inclusion criteria. Subgroup analyses revealed that compared with the drug group, the drug and acupuncture group had a greater positive effect on the total effective rate (TER) (nine trials, n=427; RR=1.20; 95% CI 1.08 to 1.32; P-heterogeneity=0.20, $I^2$=28%, p=0.0004) and the recovery rate (RCR) (six trials, n=294; RR=1.61; 95% CI 1.30 to 1.98; P-heterogeneity=0.29, $I^2$=19%, p<0.0001) of PGS after abdominal surgery. However, acupuncture showed no significant advantages in terms of the TER after thoracic surgery (one trial, p=0.13) or thoracic/abdominal surgery-related PGS (two trials, n=115; RR=1.18; 95% CI 0.89 to 1.57; P-heterogeneity=0.08, $I^2$=67%, p=0.24) and the RCR after thoracic/abdominal surgery (two trials, n=115; RR=1.40; 95% CI 0.97 to 2.01; P-heterogeneity=0.96, $I^2$=0%, p=0.07). The quality of evidence for TER and RCR was moderate certainty. Only one study reported an acupuncture-related adverse event, in the form of mild local subcutaneous haemorrhage and pain that recovered spontaneously. TSA indicated that outcomes reached a necessary effect size except for clinical symptom score.

**Conclusion** Based on subgroup analysis, compared with the drug treatment, acupuncture combined drug has significant advantages in the treatment of PGS associated with abdominal surgery, but not with thoracic surgery.

**PROSPERO registration number** CRD42022299189.

## STRENGTHS AND LIMITATIONS OF THIS STUDY

⇒ This systematic review and meta-analysis was performed according to the Preferred Reporting Items for Systematic Reviews and Meta-Analyses statement to ensure the reproducibility of the study.

⇒ A comprehensive search strategy was used to search through a large number of databases and registration websites, and grey literatures were searched in the study.

⇒ The quality of evidence for each outcome was evaluated using Grading of Recommendations Assessment, Development and Evaluation, the robustness of conclusions was examined through trial sequential analysis, and publication bias was assessed with Egger's test using Stata V.16 software.

⇒ Outcomes were analysed by subgroups according to surgical site, acupuncture stimulation method and intervention course.

⇒ The included outcomes were immediate postintervention time point data, leading to a lack of analysis of the long-term effects of the intervention in this study.

## INTRODUCTION

Postsurgical gastroparesis syndrome (PGS) is a gastric motility disease defined by delayed gastric emptying after eating without pyloric stenosis or another mechanical obstruction within 3 months postoperatively, with symptoms including nausea, vomiting, early satiety, postprandial fullness and bloating.[1–3] PGS is the second most common type of gastroparesis after diabetes gastroparesis, and it often occurs after abdominal surgery,[2] with an incidence rate of 0.4%–5.0% after gastric surgery.[4 5] In addition, PGS has been reported to occur after thoracic surgery, such

as oesophageal surgery,[6 7] lung surgery[8 9] and heart transplantation,[10 11] which can negatively affect patients' postoperative quality of life.

Treatment of PGS consists of dietary modification, gastrointestinal decompression (GD), fluid and electrolyte acid-base balance, nutritional support, psychological therapies and prokinetic and antiemetic medication, but with limited benefit.[1] Additional therapeutic options such as pyloric botulinum toxin injections, gastric electrical stimulation (GES), peroral pyloromyotomy (POP) and gastric peroral endoscopic myotomy (G-POEM) are considered for PGS when medication fails, but all these approaches are invasive and have limited efficacy.[12 13] The gastrointestinal total symptom score in patients treated with GES was reduced by 48% in Europe and 38% in the USA.[14] The effect of modern medical treatment on PGS is unsatisfying. Acupuncture is a traditional medical treatment that has been applied in many therapeutic areas and is getting more attention.[15] Acupuncture, or a combination of acupuncture and drugs, has been reported to have some advantages in the treatment of PGS in recent clinical trials.[16 17]

Invasive acupuncture, which involves piercing one or more needles into specific acupoints on the body surface, is a traditional Chinese medicine treatment method that is distinct from other local stimulation therapies such as transcutaneous electrical nerve stimulation.[18] Several studies have documented the mechanism of acupuncture for PGS as follows: (1) regulating the secretion of gastrointestinal hormones. In an animal study, Yu et al[19] found that acupuncturing at Zhongwan (CV12) and Zusanli (ST36) promoted the levels of motilin (MTL), gastrin (GAS), ghrelin and growth factor 1 in serum of rats to treat gastroparesis. In a clinical trial, Xing et al[20] found that electroacupuncture improved gastrointestinal function and upregulated serum MTL and GAS levels in patients with acute gastrointestinal injury after 7 days of intervention. In another clinical trial, Chen et al[21] found that acupuncture combined with domperidone more significantly upregulated the contents of MTL and GAS in plasma of patients with gastroparesis compared with domperidone alone. In a randomised controlled trial (RCT), Güçel et al[22] found that after several weeks of acupuncture treatment, patients' plasma ghrelin was upregulated and gastrointestinal function was improved. (2) Stimulating the vagus nerve. The pathogenesis of PGS may be related to vagus nerve injury or compression and impaired gastric muscle regulation.[1] In animal studies, Yang et al[23] and Noguchi[24] found that electroacupuncturing at ST36 on the hind limb stimulated the vagus nerve, which attenuated motility dysfunction in intestinal muscularis and modulated gastrointestinal motility. In a clinical trial, Liu et al[25] found that the improvements in volunteers' fullness, satiety, bloating, nausea, vomiting and gastric slow waves were closely related to the stimulation of the vagus nerve by acupuncture. (3) Reducing damage to interstitial cells of Cajal (ICCs). ICCs are considered pacemaker cells in the gastrointestinal tract

that transmit electrical signals for gastric emptying.[1] In an animal study, Tian et al[26] found that electroacupuncturing at ST36 protected ICCs by regulating macrophages to improve gastric motility. In a clinical trial, Song et al[27] found that the improvement of gastroparesis by acupuncture could be related to reducing the damage to ICCs.

Several previous clinical trials have observed therapeutic effects of acupuncture on PGS, such as improving symptoms of nausea, vomiting, early satiety and bloating; reducing the time to resumption of diet for the first time (RDT); promoting gastric emptying; downgrading gastric juice volume (GJV) and upgrading MTL and GAS in serum.[17 28 29] Further clinical trials are in progress.[30 31]

The lack of convincing evidence for the efficacy of acupuncture in the treatment of PGS due to the small sample size and different trial protocols has hampered its clinical application. This study systematically reviewed RCTs of acupuncture for PGS by meta-analysis to provide evidence for the clinical application of acupuncture.

## METHODS
### Study registration
The protocol for this systematic review and meta-analysis was registered in PROSPERO, and the study was performed according to the Preferred Reporting Items for Systematic Reviews and Meta-Analyses (PRISMA) statement as shown in online supplemental table S1.[32]

### Data sources and search strategy for study registration
We searched, with no language restrictions, 12 databases (PubMed, Embase, Cochrane Library Cochrane Central Register of Controlled Trials (CENTRAL), Medline (Ovid) (from 1946), Web of Science, EBSCO, Scopus, Open Grey, China National Knowledge Infrastructure (CNKI), Wanfang Database, Chinese Scientific Journals Database (VIP) and China Biology Medicine disc (CBM)) and three registration websites (WHO International Clinical Trials Registry Platform (ICTRP), ClinicalTrials.gov, and Chinese Clinical Trial Registry (ChiCTR)) from their creation to September 2022 for RCTs. Search strategies are detailed in online supplemental table S2. We adapted search strategy appropriately for each database. In addition, we screened citations of the included articles manually to avoid search omissions.

### Study selection
We screened titles and abstracts of articles using Endnote V.X7 software, and the full texts were screened according to the selection criteria of the study.

Inclusion criteria: (1) Participants of the trails were diagnosed with PGS.[4] There are no standardised criteria for PGS diagnosis, so we adopted the diagnostic criteria formulated by Professor Qin's team at Zhongshan Hospital, Fudan University, Shanghai, China, after researching the relevant documents.[1 3 33–35] (2) The control group of the trial was treated with prokinetic drugs in addition to the routine treatment. (3) The experimental group of the

study was treated with acupuncture on the basis of the control group. (4) Acupuncture was defined as the penetration of a filiform needle invasively into the skin with or without electrical or moxibustion stimulation. (5) The trial was an RCT with no blinding restrictions.

Exclusion criteria: (1) duplicated publications. (2) The experimental group of the study used other Chinese medical treatments in addition to acupuncture, including but not limited to oral herbs, non-invasive external therapies such as moxibustion without acupuncture, acupressure, acupoint application and herbal enema. (3) Missing or uncleared data and an inability to contact the author for further details. (4) Animal or cell experiments. (5) Reviews.

Two reviewers screened and selected articles independently. Disagreements between the reviewers were resolved by discussion with a third reviewer.

### Outcome assessment

The primary outcomes of the study were the total effective rate (TER) and the recovery rate (RCR). In the trials included in this meta-analysis, the evaluation criteria for TER and RCR of PGS were centred on these proposed by the China Association of Chinese Medicine, which are the combination of objective indicators (GJV, gastrointestinal radiography) and subjective indicators (whether nausea, vomiting and bloating disappear).[36] TER refers to the proportion of recovered and effective people in all participants. RCR refers to the proportion of recovered people in all participants. The secondary outcomes were serum MTL, serum GAS, GJV, days to gastrointestinal motility recovery (GMR), days to GD, days to RDT and clinical symptom score (CSS).

### Data extraction and management

We extracted and sorted relevant data of the included studies using Microsoft Excel 2019 and displayed them in a table. The data included the first author, year of publication, trial duration, number and age of patients in each group, surgical site, treatment methods and courses of intervention and control groups, acupoint and outcomes. Dichotomous data were the number of events and the number of participants per group; continuous data were presented as mean, SD and number of participants per group.

Data extraction and management were completed by two reviewers independently and a third reviewer joined the discussion and made a decision when disagreements arose between reviewers.

### Assessment of risk of bias and quality of evidence

Two reviewers independently evaluated the risk bias of the included studies based on the Cochrane Handbook, which contains seven items: (1) random sequence generation (selection bias), (2) allocation concealment (selection bias), (3) blinding of participants and personnel (performance bias), (4) blinding of outcome assessment (detection bias), (5) incomplete outcome data (attrition bias), (6) selective reporting (reporting bias) and (7) other biases.[37] Three levels (high risk, low risk and unclear risk) were used during the evaluation. Any disagreements during the assessment process were discussed and evaluated by a third senior professional. Review Manager V.5.3 software was used for building the summary and graph of the risk of bias. Furthermore, this study assessed the quality of evidence for each outcome and expressed it as high, medium, low or very low using Grading of Recommendations Assessment, Development and Evaluation (GRADE) pro V.3.6.[38 39]

### Statistical analysis

The systematic review and meta-analysis were completed with Review Manager V.5.3 software. During data analysis, the risk ratio (RR) with 95% CI was used for categorical data, while the mean difference (MD) with 95% CI was used for continuous data. The results were considered statistically significant when $p < 0.05$. Heterogeneity was assessed by the Cochran's $\chi^2$ test and quantified by the $I^2$ statistic test. If there was significant heterogeneity among studies ($p < 0.10$ or $I^2 > 50\%$), the random effects model was used; otherwise, the fixed effects model was used when $p \geq 0.10$ or $I^2 \leq 50\%$. Subgroup analyses were conducted according to surgical site, acupuncture stimulation method and intervention course for heterogeneity analysis. Sensitivity analysis was performed to determine the stability of the results by converting the fixed effect model and random effect model, and removing single study in turn to observe its impact on the summary results. Publication bias was assessed with Egger's test by Stata V.16 software and the funnel diagram plotted with Review Manager V.5.3 software. Trial sequential analysis (TSA) was conducted using TSA V.0.9.5.10 Beta (Copenhagen Trial Unit) software.

### Patient and public involvement statement

Neither patients nor the public were involved in the design, conduct, reporting or dissemination plans of this research.

## RESULTS
### Study selection

The flowchart of the article screening process in the meta-analysis is shown in figure 1, which was derived from the PRISMA statement.[32] A total of 2549 articles were searched, 2513 of which were from databases and 36 from registers. A total of 121 articles were selected for full-text screening by reading the titles and abstracts. Fourteen articles[40–53] were finally included after 107 articles had been excluded for reasons such as unclear diagnosis, inappropriate intervention method, study type other than RCT or unclear data. In addition, we screened 262 articles cited in the 14 included articles, and 1 of these met the inclusion criteria.[54] As a result, a total of 15 studies were included in this meta-analysis.

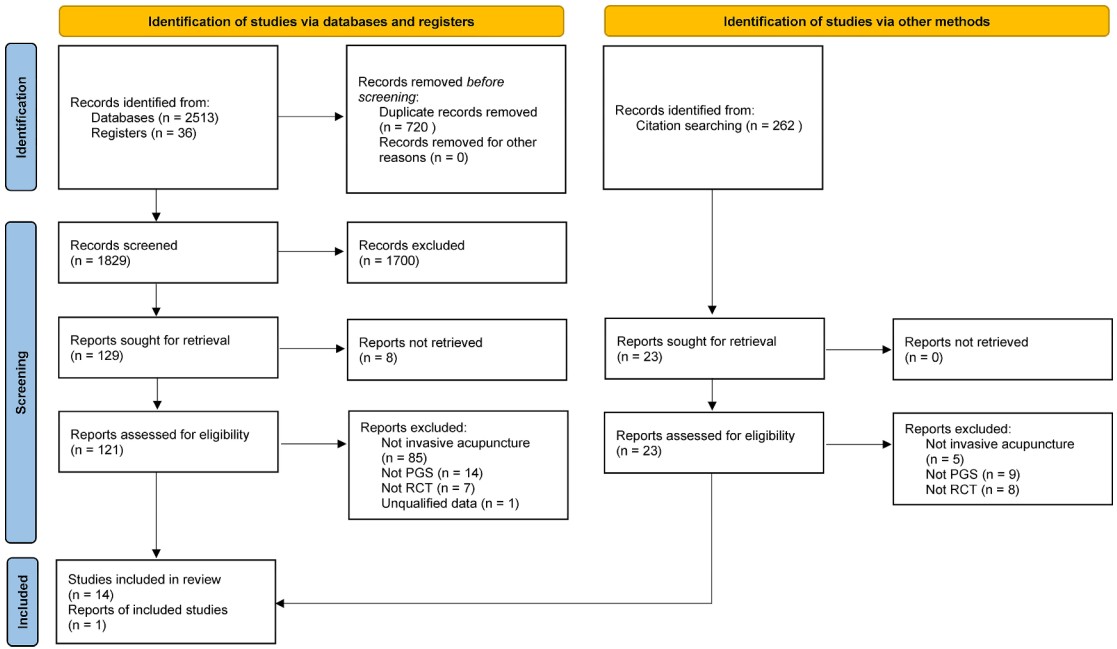

**Figure 1** PRISMA flow diagram of screening process. PGS, postsurgical gastroparesis syndrome; PRISMA, Preferred Reporting Items for Systematic Reviews and Meta-Analyses; RCT, randomised controlled trial.

## Study characteristics

The basic characteristics of the included studies are presented in online supplemental table S3. The included studies were all completed in China and included 759 patients, with sample sizes ranging from 22 to 71 participants. One of the 15 studies was a multicentre trial, and the rest were singlecentre trials. The mean age of patients was over 50 years in 10 studies, under 50 years in three studies and not mentioned in two studies. three studies included PGS after thoracic or abdominal surgery, one study included simple thoracic surgery and the rest included simple abdominal surgery. The control group (drug group) was treated with prokinetic drugs in all of the included studies, and the experimental group (drug and Ac group) was treated with a filiform needle on the basis of the control group. In eight studies, the experimental group was treated with an individual filiform needle, which was labelled manual acupuncture (MAc), while in seven studies, electrical stimulation or moxibustion stimulation was added based on a filiform needle, which was labelled stimulated MAc (SMAc). A number of studies used multiple acupoints, except for Huang 2020 who stimulated only a pair of acupoints. A total of 11 studies clearly described the treatment course, of which eight studies had a treatment course of ≤14 days and three studies had a treatment course of >14 days. The frequency of acupuncturing was once daily in 11 studies, two times per day in three studies, and between one and two times per day in one study. For primary outcomes, 12 studies reported the TER and 8 reported the RCR. Four studies received financial support from provincial projects.

## Risk of bias and study quality

All of the studies mentioned a random design, with 10 studies (66.67%) clearly reporting random sequence generation processes and only 1 study (6.67%) describing the blinding of participants. Four studies (26.67%) reported case shedding. The summary results of the risk of bias are detailed in figure 2. According to the GRADE results, the outcomes of the TER, the RCR and GSA showed moderate-certainty evidence; the MTL, GJV, GD and RDT showed low-certainty evidence and the GMR and CSS showed very low-certainty evidence (online supplemental table S4).

## Main outcomes
### Total effective rate

Twelve trials reported the TER (figure 3A). There were 302 patients in the drug and Ac group and 276 patients in the drug group. Test for the overall effect indicated that the TER of the drug and Ac group was significantly higher than that of the drug group (12 trials, n=578; RR=1.18; 95% CI 1.09 to 1.27; P-heterogeneity=0.22, $I^2$=23%, p<0.0001). The sensitivity analysis showed that the study reported by Huang 2020 was the source of instability, which may be related to acupuncturing at only one pair of acupoints (online supplemental table S5). We downgraded evidence by one level for the lack of allocation concealment and the lack of blinding. We judged the evidence to be of moderate certainty.

The results of the subgroup analysis of surgical sites revealed that the Drug and Ac Group had a statistically significant advantage in terms of the TER of PGS after abdominal surgery (9 trials, n=427; RR=1.20; 95% CI 1.08 to 1.32; P-heterogeneity=0.20, $I^2$=28%, p=0.0004), but not

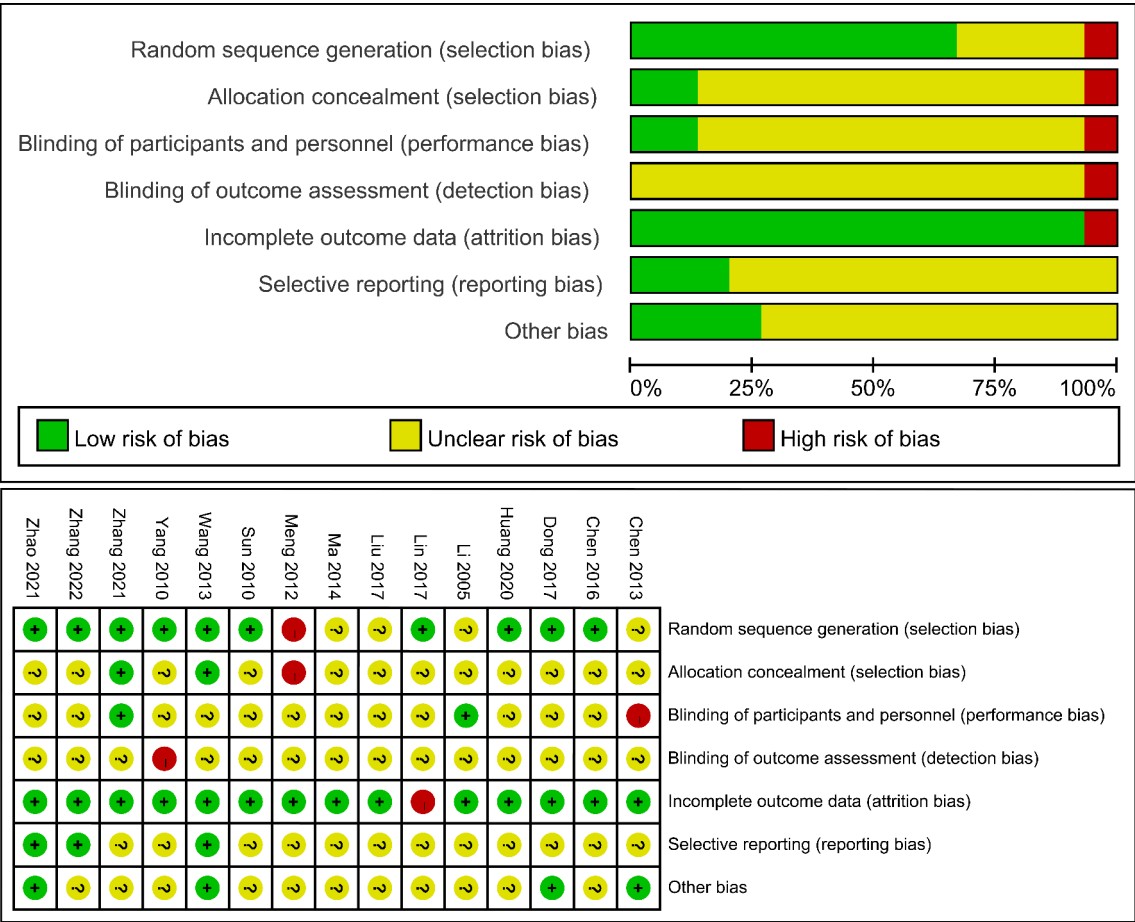

**Figure 2** Risk of bias summary and risk of bias graph.

after thoracic surgery (one trial, p=0.13) and thoracic/abdominal surgery (two trials, n=115; RR=1.18; 95% CI 0.89 to 1.57; P-heterogeneity=0.08, $I^2$=67%, p=0.24) (figure 3A). Furthermore, subgroup analysis revealed that the overall effect of the TER was not significantly related to the acupuncture course (online supplemental figure S1A) or the method of acupuncture stimulation (individual filiform needle, electrical stimulation or moxibustion stimulation) (online supplemental figure S1B).

### Recovery rate

Eight studies reported the RCR (figure 3B). There were 217 patients in the drug and Ac group and 192 patients in the drug group. Testing for the overall effect indicated that the RCR of the drug and Ac group was significantly higher than that of the drug group (eight trials, n=409; RR=1.55; 95% CI 1.29 to 1.86; P-heterogeneity=0.54, $I^2$=0%, p<0.00001). Sensitivity analysis showed that the conclusion was stable (online supplemental table S5). We downgraded evidence by one level for the lack of allocation concealment and the lack of blinding. We judged the studies as moderate-certainty evidence.

The results of the subgroup analysis of surgical sites revealed that the drug and Ac group had a statistically significant advantage in terms of the RCR of PGS after abdominal surgery (six trials, n=294; RR=1.61; 95% CI 1.30 to 1.98; P-heterogeneity=0.29, $I^2$=19%, p<0.0001), but

not after thoracic/abdominal surgery (two trials, n=115; RR=1.40; 95% CI 0.97 to 2.01; P-heterogeneity=0.96, $I^2$=0%, p=0.07) (figure 3B). The results of the subgroup analysis of acupuncture course revealed that the drug and Ac group had a statistically significant advantage in terms of the RCR of PGS when the course was ≤14 days (five trials, n=264; RR=1.68; 95% CI 1.31 to 2.16; P-heterogeneity=0.29, $I^2$=19%, p<0.0001), but when the course was >14 days (two trials, n=83; RR=1.29; 95% CI 0.95 to 1.74; P-heterogeneity=0.98, $I^2$=0%, p=0.10) (online supplemental figure S2A). The results of the subgroup analysis of the method of acupuncture stimulation revealed that the drug and Ac group had a significant advantage in terms of the RCR of PGS when treated with SMAc (six trials, n=296; RR=1.40; 95% CI 1.15 to 1.71; P-heterogeneity=0.96, $I^2$=0%, p=0.0007), but not when treated with MAc (two trials, n=113; RR=2.24; 95% CI 0.70 to 7.15; P-heterogeneity=0.03, $I^2$=79%, p=0.17) (online supplemental figure S2B).

### Secondary outcomes
#### Motilin

MTL, which has a positive effect on PGS, is a key gastrointestinal hormone that can facilitate gastrointestinal motility by raising the secretion of pepsin and acid in the stomach and causing contraction of the stomach.[55] Serum MTL levels in healthy individuals are around

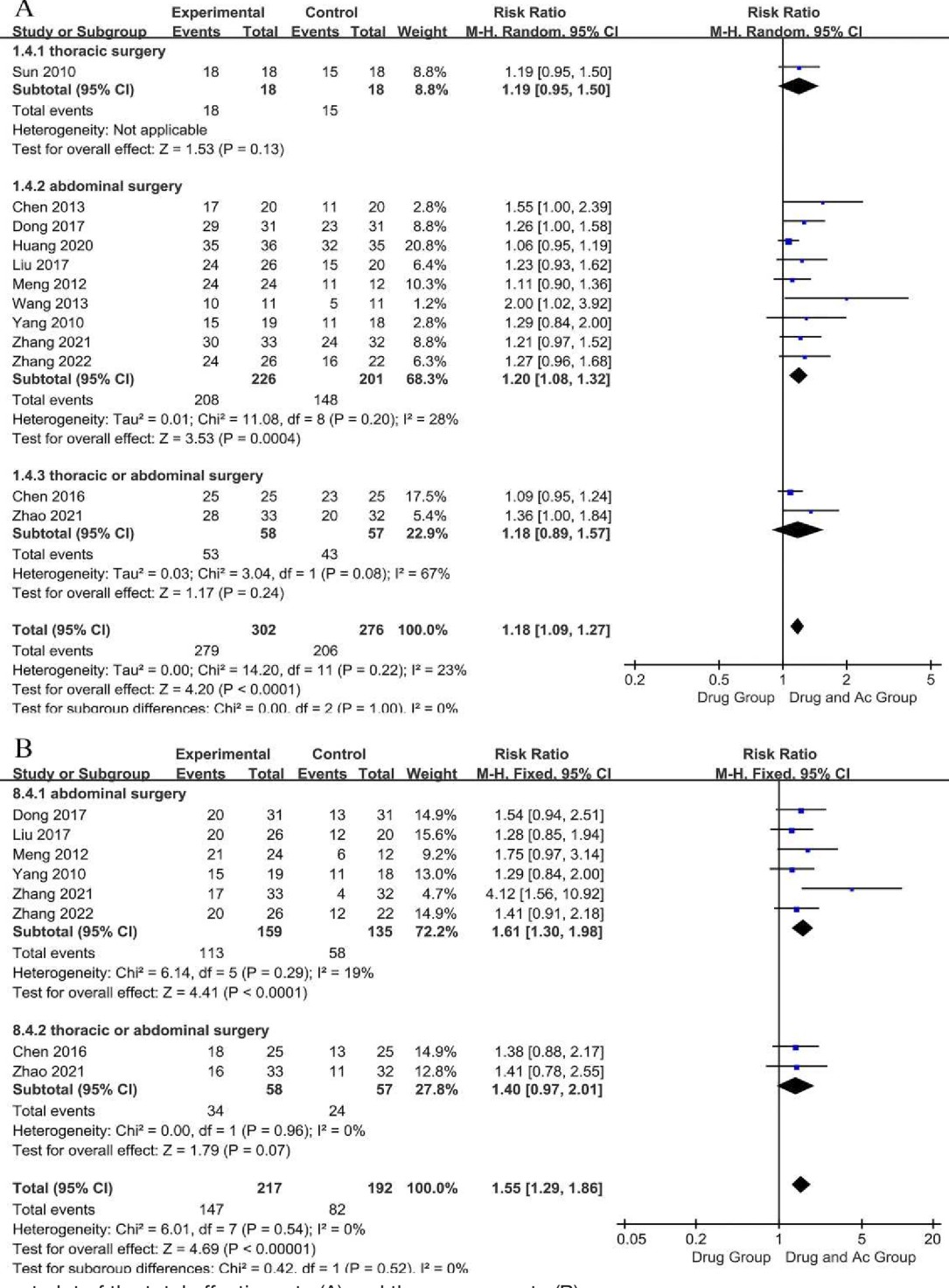

**Figure 3** Forest plot of the total effective rate (A) and the recovery rate (B).

499.02 pg/mL.[56] Five studies reported the MTL levels (figure 4A). There were 152 patients in the drug and Ac group and 152 patients in the drug group. Testing for the overall effect indicated that the MTL level of the drug and Ac group was significantly higher than that of the drug group (five trials, n=304; MD=68.15; 95% CI 42.89 to 93.40; P-heterogeneity<0.0001, $I^2$=86%, p<0.00001). The sensitivity analysis showed that the study reported by Lin[45] was the source of instability, which may be related to the high risk of bias (online supplemental table S5). We

downgraded evidence by two levels for the lack of allocation concealment, the lack of blinding and unexplained heterogeneity. We judged the studies as low-certainty evidence.

The results of the subgroup analysis of surgical sites revealed that the drug and Ac group had a statistically significant advantage in terms of the MTL levels of PGS after both abdominal surgery (three trials, n=198; MD=48.76; 95% CI 37.98 to 59.54; P-heterogeneity=0.47, $I^2$=0%, p<0.00001) and thoracic/abdominal surgery

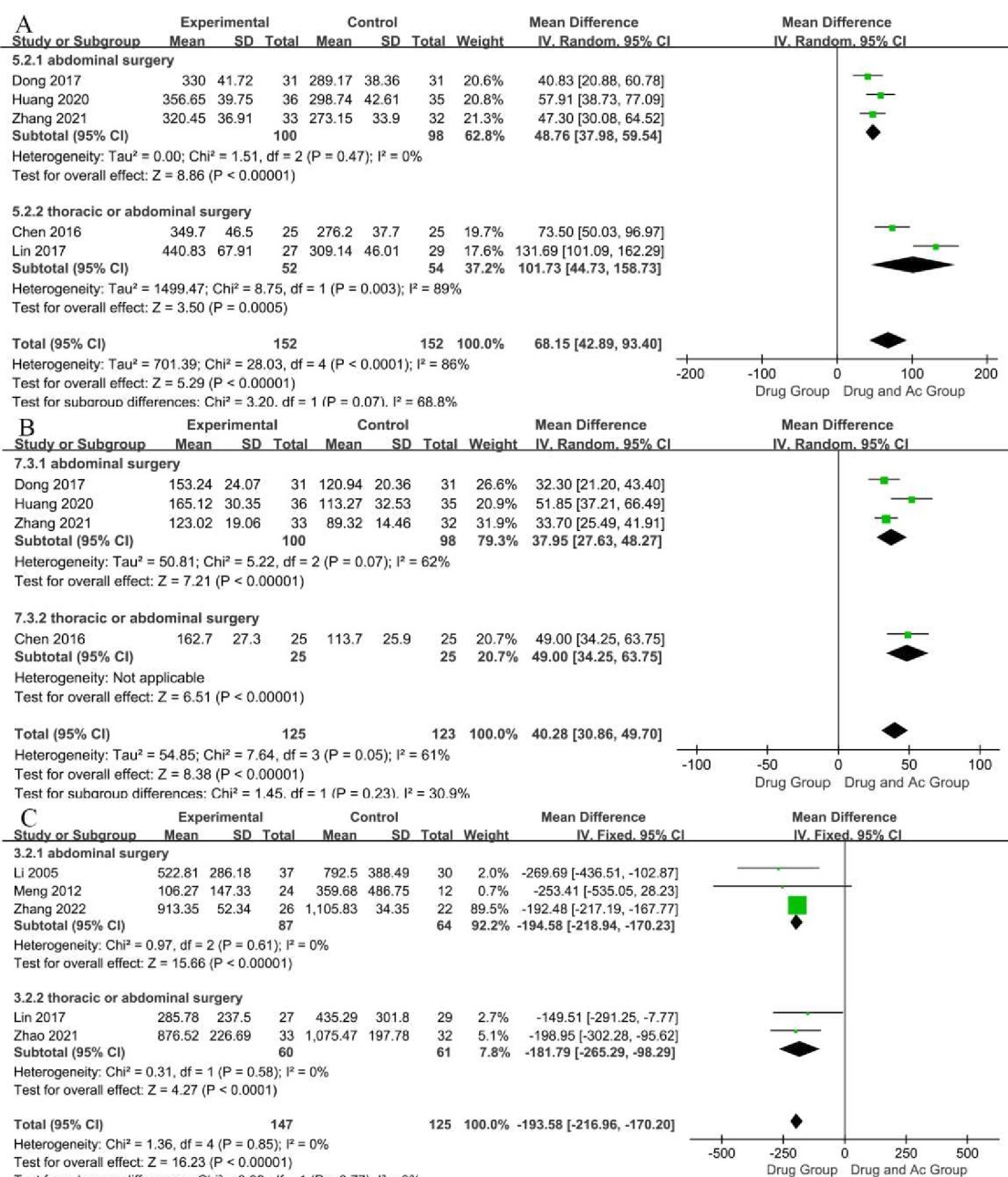

**Figure 4** Forest plot of the serum motilin (A), the serum gastrin (B) and the gastric juice volume (C).

(two trials, n=106; MD=101.73; 95% CI 44.73 to 158.73; P-heterogeneity=0.003, I²=89 %, p=0.0005), but with high heterogeneity (figure 4A).

### Gastrin

GAS is another gastrointestinal hormone that enhances gastric motility and has a positive effect on PGS.[57] Serum MTL levels in healthy individuals are around 92.95 pg/mL.[56] Four studies reported the GAS levels (figure 4B). There were 125 patients in the drug and Ac group and 123 patients in the drug group. Testing for the overall effect indicated that the GAS levels of the drug and Ac group were significantly higher than that of the drug group (four trials, n=248; MD=40.28; 95% CI 30.86 to 49.70; P-heterogeneity=0.05, I²=61%, p<0.00001).

Sensitivity analysis showed that the conclusion was stable despite high heterogeneity (online supplemental table S5). However, the cause of heterogeneity was not found, and higher quality trials may be needed for further study. We downgraded evidence by one level for the lack of allocation concealment and the lack of blinding. We judged the studies as moderate-certainty evidence.

The results of the subgroup analysis of surgical sites revealed that the drug and Ac group had a statistically significant advantage in terms of the GAS levels of PGS after both abdominal surgery (three trials, n=198; MD=37.95; 95% CI 27.63 to 48.27; P-heterogeneity=0.07, I²=62 %, p<0.00001) and thoracic/abdominal surgery (one trial, p<0.00001) (figure 4B).

## Gastric juice volume

The volume of gastric juice secreted by normal adults is 1.5~2.5 L/day.[58] Gastric juice accumulates in the stomach with gastric hypomotility. Higher levels of GJV indicate lower gastric motility. Patients' GJV levels in individuals without surgeries are around 45.2 mL/day.[59] Five studies reported the GJV (figure 4C). There were 147 patients in the drug and Ac group and 125 patients in the drug group. Test for the overall effect indicated that the GJV of the drug and Ac group was significantly lower than that of the drug group (five trials, n=272; MD=−193.58; 95% CI −216.96 to −170.20; P-heterogeneity=0.85, I²=0%, p<0.00001). Sensitivity analysis showed that the conclusion was stable (online supplemental table S5). We downgraded evidence by two levels for the lack of allocation concealment, the lack of blinding and incomplete accounting of patients and outcome events. We judged the studies as low-certainty evidence.

The results of the subgroup analysis revealed that the advantage of the drug and Ac group in regulating GJV was not affected by the surgical site (figure 4C) or acupuncture method (online supplemental figure S3).

## Days to GMR

GMR is an indicator of the effectiveness of PGS treatment, with smaller values indicating better efficacy. Six studies reported the days to GMR (figure 5A). There were 186 patients in the drug and Ac group and 181 patients in the drug group. Testing for the overall effect indicated that there were significantly fewer days to GMR in the drug and Ac group than in the drug group (six trials, n=367; MD=−4.87; 95% CI −7.80 to −1.94; P-heterogeneity< 0.00001, I²=97%, p=0.001). Sensitivity analysis showed that the conclusion was stable despite high heterogeneity (online supplemental table S5). We downgraded evidence by three levels for the lack of allocation concealment, the lack of blinding, incomplete accounting of patients and outcome events and unexplained heterogeneity. We judged the studies as very low-certainty evidence.

The results of the subgroup analysis revealed that the advantage of the drug and Ac group in regulating GMR was not affected by surgical site (figure 5A).

## Days to GD

GD is an indicator of the effectiveness of PGS treatment, with smaller values indicating better efficacy. Four studies reported the days to GD (figure 5B). There were 112 patients in the drug and Ac group and 110 patients in the drug group. Testing for the overall effect indicated that there were significantly fewer days to GD in the drug and Ac group compared with the drug group (four trials, n=222; MD=−3.47; 95% CI −5.74 to −1.21; P-heterogeneity<0.00001, I²=94%, p=0.003). Sensitivity analysis showed that the study reported by Ma[47] was the source of instability (online supplemental table S5), this might be related to the fact that some of the participants in this trial were treated with sedatives, as sedatives can affect vagal activity.[60] We downgraded evidence by two levels for

the lack of allocation concealment, the lack of blinding and unexplained heterogeneity. We judged the studies as low-certainty evidence.

## Days to RDT

RDT is an indicator of the effectiveness of PGS treatment, with smaller values indicating better efficacy. Two studies reported the days to RDT (figure 5C). There were 64 patients in the drug and Ac group and 63 patients in the drug group. Testing for the overall effect indicated that there were significantly fewer days to RDT in the drug and Ac group than in the drug group (two trials, n=127; MD=−2.68; 95% CI −4.23 to −1.12; P-heterogeneity= 0.03, I²=78 %, p=0.0007). Sensitivity analysis showed that the conclusion was stable despite high heterogeneity (online supplemental table S5). The reliability of this conclusion was low due to the lack of data. We downgraded evidence by two levels for unexplained heterogeneity and the small sample size. We judged the studies as low-certainty evidence.

## Clinical symptom score

CSS is an indicator of the effectiveness of PGS treatment, with smaller scores indicating better efficacy. Three studies reported the CSS, and they were scored with the same criteria (figure 5D). There were 62 patients in the drug and Ac group and 58 patients in the drug group. Testing for the overall effect indicated that the CSS of the drug and Ac group was lower than that of the drug group, but the difference was not statistically significant (three trials, n=120; MD=−1.31; 95% CI −3.24 to 0.61; P-heterogeneity<0.00001, I²=100%, p=0.18). We downgraded evidence by four levels for the lack of allocation concealment, the lack of blinding and incomplete accounting of patients and outcome events, unexplained heterogeneity and the small sample size. We judged the studies as very low-certainty evidence.

The results of the subgroup analysis of surgical site revealed that the drug and Ac group had no statistically significant advantage in terms of the CSS of PGS after either abdominal surgery (two trials, n=70; MD=−0.17; 95% CI −0.43 to 0.08; P-heterogeneity=0.01, I²=85%, p=0.18) or thoracic/abdominal surgery (one trial, <0.00001 95% CI) (figure 5D).

In addition, two studies reported the CSS with two different criteria, and their conclusions were different (Chen 2013: n=40; MD=−1.05; 95% CI −3.33 to 1.23; p=0.37;[45] n=56; MD=−0.34; 95% CI −0.45 to 0.23; p<0.00001).

## Safety analysis

Two studies described the safety of the trials. Dong assessed nutritional indicators such as albumin, haemoglobin and transferrin before and after the intervention. The results showed that all three indicators were significantly higher in the experimental and control groups at the end of the intervention compared with the preintervention period, and the three indicators in the experimental

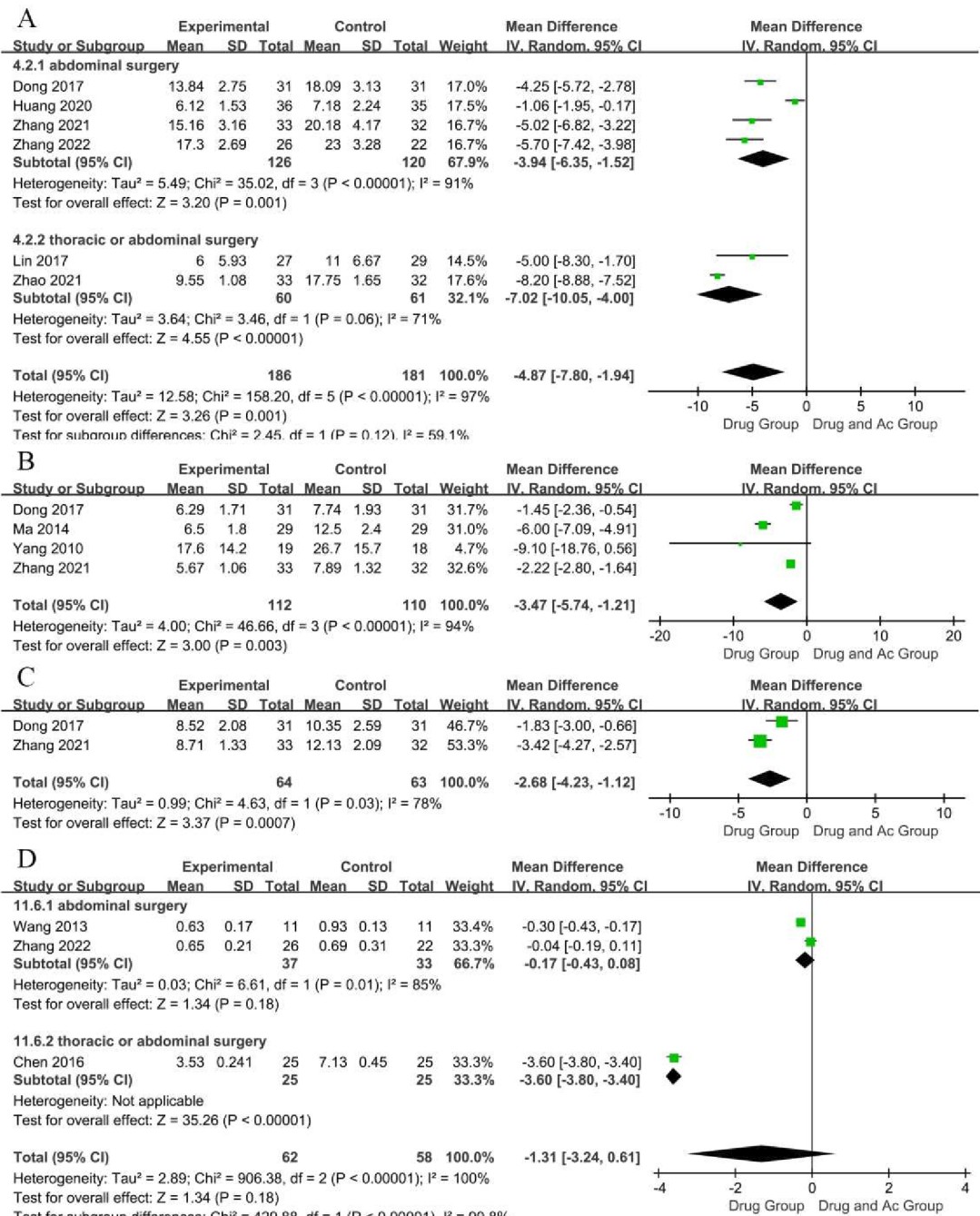

**Figure 5** Forest plot of days of gastrointestinal motility recovery (A), days of gastrointestinal decompression (B), days to resumption of diet for the first time (C) and clinical symptom score (D).

group were significantly higher than those in the control group, with statistically significant differences (p<0.05). Zhao recorded adverse events that included subcutaneous haemorrhage and pain associated with acupuncture in the experimental group, but they were all mild and recovered without special treatment. There were no acupuncture-related serious adverse events in the experimental and control groups.

### Publication bias
The funnel plot of the TER was asymmetric (online supplemental figure S4), although Egger's test showed p=0.399 > 0.05 (online supplemental table S6), indicating that there was risk of publication bias in the included studies.

### Trial sequential analysis
TSA showed that the cumulative Z-curve of the TER and the RCR surpassed the trial sequential monitoring boundary and the conventional test boundary. The cumulative Z-curve of the MTL, GAS, GJV, GMR, GD, RDT surpassed the trial sequential monitoring boundary, the conventional test boundary and the required information size (RIS). The results suggested that these conclusions

**Table 1** Summary of frequencies of the applicated acupoints (top nine)

| Number | Acupoint | Alphabetic code | Frequency |
|---|---|---|---|
| 1 | Zusanli | ST36 | 15 |
| 2 | Zhongwan | CV12 | 12 |
| 3 | Neiguan | PC6 | 11 |
| 4 | Shangjuxu | ST37 | 8 |
| 5 | Tianshu | ST25 | 6 |
| 6 | Sanyinjiao | SP6 | 6 |
| 7 | Gongsun | SP4 | 5 |
| 8 | Qihai | CV6 | 4 |
| 9 | Yinlingquan | SP9 | 3 |

were robust and no further trials are necessary to verify these conclusions. However, the cumulative Z-curve of the CSS failed to surpassed the trial sequential monitoring boundary or the RIS, suggesting that the conclusion was not robust and further trials are necessary to verify it (online supplemental figure S5).

### Frequency analysis of acupoints
The 15 studies involved 26 acupoints with a frequency of 88. The acupoints that were applied ≥3 times are listed in table 1, according to the application frequency from high to low, with the top three being ST36, CV12 and PC6.

### DISCUSSION
This study aimed to evaluate the efficacy of acupuncture in the treatment of PGS. A total of 15 studies containing 759 patients were included after systematic searching and screening of the literature. Our meta-analysis suggested that, overall, acupuncture combined with prokinetic drugs had a more positive efficacy on the TER, RCR, MTL, GAS, GJV, days to GMR, days to GD and days to RDT in PGS compared with prokinetic drugs alone. Although there was a trend towards the positive efficacy of acupuncture combined with drugs on CSS, the difference was not statistically significant. The heterogeneity of the included CSS-related trials was significant, which may be because the CSS is a subjective indicator and there were possibilities of bias. At present, there are too few relevant trials to analyse the reasons of the heterogeneity. The results of the subgroup analysis of the main outcomes suggested that the advantage of the TER of acupuncture combined with prokinetic drugs for PGS was not influenced by the course of acupuncture, or the method of acupuncture stimulation; however, the RCR of acupuncture for PGS showed a statistically significant advantage only for the short course. It is noteworthy that acupuncture combined with prokinetic drugs had no obvious advantage in terms of the TER and RCR of PGS after thoracic surgery and thoracic/abdominal surgery but significantly improved the TER and RCR of PGS after abdominal surgery. In addition, the therapeutic advantage in terms of secondary outcomes in the

group treated with acupuncture combined with prokinetic drugs was not affected by acupuncture course, method, frequency or surgical site, although there was heterogeneity. As for safety analysis, only one study reported mild adverse events associated with acupuncture, and another study reported the levels of albumin, haemoglobin and transferrin in serum. Evidence was generally insufficient, and more research is needed to assess the safety of acupuncture in patients with PGS. The quality of evidence was deemed to be low due to the limitations such as the lack of allocation concealment, lack of blinding, incomplete accounting of patients and outcome events, unexplained heterogeneity and the small sample size. TSA suggested that the conclusions, except for CSS, were robust and no further trials are necessary to verify these conclusions.

At present, prokinetic drugs are the most commonly used in PGS medication. Metoclopramide, a D2 receptor antagonist, is used with some 5-HT4 receptor agonists at the same time. As the only drug approved by Food and Drug Administration (FDA) for gastroparesis, metoclopramide can effectively alleviate PGS-related nausea, vomiting and other symptoms of the digestive system.[61] 5-HT4 agonists such as cisapride are frequently and early adopted in PGS, and mosapride has also been used in gastroparesis.[62–65] Prucalopride is a recently discovered effective drug that accelerates gastric emptying without cardiac side effects and has been initially reported to relieve symptoms in patients with idiopathic gastroparesis or diabetic gastroparesis,[66] but high-quality studies in PGS are lacking. Macrolide antibiotics such as erythromycin are agonists of GAS receptors, which induce GAS secretion in the proximal part of the digestive tract and accelerate gastric emptying. However, they face problems such as rapid drug resistance, and their clinical benefit is controversial.[67 68]

The results showed that acupuncture combined with prokinetic drugs for PGS was able to significantly improve the efficacy of the treatment and digestive symptoms of patients. Meanwhile, acupuncture is less traumatic compared with endoscopic and surgical treatments, such as pyloric botulinum toxin injections, GES, POP and G-POEM.[69]

Among the 15 studies included in this meta-analysis, ST36 on the ST meridian, CV12 on the CV meridian and PC6 on the PC meridian were the most frequently adopted acupoints, and all three acupoints play an important role in regulating stomach function. It has been reported that acupuncturing at ST36 can contract muscles,[70] regulate gastric rhythm and improve gastric emptying delay by stimulating the parasympathetic nerves to treat gastroparesis.[71] Further study indicated that electroacupuncturing at ST36 could modulate gastric motility and improve gastrointestinal symptoms through M2/3 and β1/2 receptor-mediated vagal and sympathetic reflex.[72] Electroacupuncturing at CV12 can regulate gastric emptying and demonstrate beneficial effects on nausea and vomiting, which may be mediated through the inhibition of 5-HT secretion in the duodenum and activity of the nucleus of the solitary tract.[73 74] Stimulating at PC6 has an effect on gastrointestinal symptoms and prevents postoperative nausea and vomiting.[75–77] Electroacupuncturing at PC6 reduces the inhibition of efferent vagus nerve motor fibres by inhibiting GABA transmission to dorsal motor nucleus of the vagus, thereby promoting efferent vagus nerve activity and increasing gastric motility.[78] In clinical practice, ST36, CV12 and PC6 are often stimulated in combination to improve digestive symptoms and have been proven to be effective.[79]

In general, the meta-analysis of previous studies of acupuncture for PGS showed that acupuncture combined with prokinetic drugs improved the TER and RCR of PGS. However, more evidences were needed on the relationship between surgical site and the TER and the RCR and the relationship between the RCR and the acupuncture course or the method of acupuncture stimulation. In addition, acupuncture can up-regulate the levels of MTL and GAS in serum, decrease GJV, accelerate GMR and shorten days to GD and RDT.

Clinical trials on acupuncture for PGS are generally single-centre studies based on the small sample size, so the efficacy remains unclear. This meta-analysis provided an extensive search and analysis for RCTs about PGS to comprehensively evaluate the therapeutic effect of acupuncture. We tried to provide an approach for clinicians and researchers to treat patients with PGS. In addition, we counted the acupoints involved in the included trials to provide reference to select acupoints and formulate treatment plans.

There has not been any meta-analysis of acupuncture for PGS after thoracic or abdominal surgery. We conducted subgroup analysis according to surgical sites and found differences in efficacy. Besides, we excluded literatures with other Chinese medical treatments such as oral herbs, acupressure and herbal enema and focused on acupuncture to make the interventions in different trials as consistent as possible. Furthermore, we assessed the quality of evidence using GRADE and evaluated the robustness of conclusions with TSA to make the methodology more rigorous.

However, this meta-analysis has some limitations. First, as mentioned, 759 participants were included in this study, which is a small sample size and there is a risk of bias. Second, all of the included studies were completed in China, and there may be racial and geographical variability in the meta-analysis. Third, the types, doses and methods of prokinetic drugs used in the studies were inconsistent. Fourth, in the analysis of CSS, the difference between experimental group and control group was not statistically significant and the heterogeneity was very large, more high-quality studies are needed. Fifth, there was a risk of publication bias in this study. Sixth, the grades of PGS were not mentioned in the included studies, which may be a key factor influencing the therapeutic effect of acupuncture on PGS. Seventh, the thoracic surgeries included in this meta-analysis were all oesophageal cancer surgeries, which led to limited data, and further in-depth studies should demonstrate the efficacy of acupuncture for PGS after other types of thoracic surgery reliably. Eighth, only two of the included studies reported safety data, and more trials are demanded. Finally, although TSA indicated that the conclusions were robust, bias risk existed in the included studies and more high-quality studies are needed to substantiate them.

## CONCLUSIONS

This meta-analysis demonstrated that, compared with the use of prokinetic drugs alone, acupuncture with prokinetic drugs for PGS associated with abdominal surgery is more advantageous. It is worthy of clinical promotion because of the improved efficacy. However, the efficacy of acupuncture for PGS associated with thoracic surgery is unclear. Due to the limitations of the included studies, high quality trials with rigorous design and large sample sizes are needed to further confirm the efficacy and safety of acupuncture for PGS.

**Contributors** JH proposed the original concept and ZF designed the study. YQ and XQ screened the abstract and full text. XQ and JX extracted the data. YW and JX evaluated the risks of bias. ZF and YQ carried out the outcomes and wrote the manuscript. YH revised the manuscript and arbitrated disagreements. WN guided the meta-analysis. JH acted as guarantor.

**Funding** This work was supported by Key Research Project of Beijing University of Chinese Medicine (Grant Number 2020-JYB-ZDGG-143-1), National Natural Science Foundation of China (Grant Number 82074545), Hospital-level Cultivation Project of the Third Affiliated Hospital of Beijing University of Chinese Medicine (Grant Number BZYSY-2022-PYQN-05), Project of Famous Doctor Inheritance Workstation of Zhang Daizhao, '3+3' Project of Beijing Traditional Chinese Medicine Inheritance (Grant Number 2007-SZ-C-12).

**Competing interests** None declared.

**Patient and public involvement** Patients and/or the public were not involved in the design, or conduct, or reporting, or dissemination plans of this research.

**Patient consent for publication** Not applicable.

**Ethics approval** Not applicable.

**Provenance and peer review** Not commissioned; externally peer reviewed.

**Data availability statement** All data relevant to the study are included in the article or uploaded as supplementary information.

**ORCID iDs**
Zhenjia Fan http://orcid.org/0000-0002-9827-3298
Yuqin Qiu http://orcid.org/0000-0001-7453-373X

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
