## [Reviewer comments · BMJ Open]

ARTICLE DETAILS

TITLE (PROVISIONAL)	Invasive acupuncture for gastroparesis after thoracic or abdominal surgery: a systematic review and meta-analysis
AUTHORS	Fan, Zhenjia; Qiu, Yuqin; Qi, Xuewei; Xu, Jingnan; Wan, Yuxiang; Hao, Yingxu; Niu, Wenquan; Huang, Jinchang

VERSION 1 – REVIEW

REVIEWER	He, Jing Guangzhou Women and Children's Medical Center, Department of Pediatric Surgery
REVIEW RETURNED	28-Dec-2022

GENERAL COMMENTS	In the current study, the authors investigate the efficacy and safety of acupuncture in the treatment of postsurgical gastroparesis syndrome (PGS) after thoracic or abdominal surgery in the form of meta-analysis by including 12 databases and 3 trials. They including 15 studies with 759 patients and found that acupuncture had a greater positive effect on the TER and RR. They declared acupuncture has a significant advantage in the treatment of PGS associated with abdominal surgery, while it hasn't in PGS associated with thoracic surgery. I have some comments that should be addressed by the authors. 1.The authors should provide the full names for the abbreviations the first time they appeared.2.The authors should check their manuscript carefully and make consistence of the style.3. Clinical importance of this meta-analysis should be emphasized. What information have the authors provided for clinicians and researchers?4. Is there any meta-analysis focused on this topic, the author should discuss the priority of this one. More discussions were needed.5. Trial sequential analysis should be performed to see if more investigations were needed. (Fu W, et al. Oncotarget. 2017; 8: 9806-9822.)
--

REVIEWER	Bhat, Sameer The University of Auckland
REVIEW RETURNED	04-Feb-2023

GENERAL COMMENTS	Fan and colleagues performed a systematic review and meta-analysis to assess the efficacy and safety of invasive acupuncture for treating postsurgical gastroparesis syndrome after thoracic or abdominal surgery. Overall, the study is well researched and
--

	written, with clear objectives and a well-constructed analysis. However, I have a few comments / concerns:  - The authors have defined "total effective rate" and "recovery rate" as their primary study outcomes. It is unclear what these relate to, and whether the definitions are homogeneous across the 12 and 8 trials, respectively, which assess these outcomes. Please comment and discuss. - The authors have concluded that acupuncture may be safe to perform due to improvements in clinical symptoms and lack of demonstrated serious adverse events. I would suggest that a more cautious and conservative statement is used (or this is removed from the manuscript conclusion entirely) given that adverse outcomes associated with acupuncture were reported in only two trials. Furthermore, heterogeneity among trials which used the clinical symptom score was very large (I²=100%), despite the authors' subgroup analysis based on the site of surgery.
--	--

REVIEWER	Noma, Hisashi The Institute of Statistical Mathematics, Department of Data Science
REVIEW RETURNED	18-Feb-2023

GENERAL COMMENTS	I am a statistical reviewer. There are several relevant concerns for the statistical methods.  1. The authors used "RR" as abbreviations of "risk ratio" and "recovery rate". It is confusable. 2. The authors determined the uses of FE or RE models by whether I² exceeds 50% or not. However, I² statistic is solely a summary statistic and it involves uncertainty. It is not an adequate reason. Please re-check the Cochrane Handbook for this issue. 3. Line 145: They noted "Subgroup studies were conducted", but I guess it would be "subgroup analyses". 4. In Statistical Analyses section, they should mention they used the Egger test for the publication bias test. 5. Although the Egger test was not significant, the funnel plot in S-Figure 4 was substantially asymmetric. It might indicate publication biases, and it should be at least mentioned as a limitation of this study in Discussion section. 6. The English should be checked by native English speakers.
---

VERSION 1 – AUTHOR RESPONSE

Reviewer: 1

Dr. Jing He, Guangzhou Women and Children's Medical Center

Comments to the Author:

In the current study, the authors investigate the efficacy and safety of acupuncture in the treatment of postsurgical gastroparesis syndrome (PGS) after thoracic or abdominal surgery in the form of meta-analysis by including 12 databases and 3 trials. They including 15 studies with 759 patients and found that acupuncture had a greater positive effect on the TER and RR. They declared acupuncture has a significant advantage in the treatment of PGS associated with abdominal surgery, while it hasn't in PGS associated with thoracic surgery.

I have some comments that should be addressed by the authors.

Comment 1. The authors should provide the full names for the abbreviations the first time they appeared.

Response 1. Thank you for your comments, and we have carefully revised our manuscript to make sure that full names for the abbreviations are provided at the first time they appeared.

Comment 2. The authors should check their manuscript carefully and make consistence of the style.
Response 2. We apologize for this confusion generated by the previous version of the manuscript. We have checked it carefully and kept the style consistent and invited native English speakers for language corrections to make it readable. We really hope that the readability of our manuscript has been substantially improved.

Comment 3. Clinical importance of this meta-analysis should be emphasized. What information have the authors provided for clinicians and researchers?

Response 3. We have emphasized the clinical importance of this meta-analysis was providing an extensive search and analysis for RCTs about postsurgical gastroparesis syndrome (PGS) to comprehensively evaluated the therapeutic effect of acupuncture [Line 463]. We have supplemented its implications to clinicians and researchers in the Discussion section such as provided an approach to treat patients with PGS, and provided references to select acupoints and formulate treatment plans [Line 466].

Comment 4. Is there any meta-analysis focused on this topic, the author should discuss the priority of this one. More discussions were needed.

Response 4. We found that there was no meta-analysis focused on acupuncture for PGS after thoracic or abdominal surgery through a database search. We have discussed the priorities of the meta-analysis such as performed subgroup analysis according to surgical site, excluded other TCM treatments during literature screening to make interventions as consistent as possible, assessed the quality of evidence using GRADE, and evaluated outcomes with TSA [Line 469].

Comment 5. Trial sequential analysis should be performed to see if more investigations were needed. (Fu W, et al. *Oncotarget*. 2017; 8: 9806-9822.)

Response 5. Thank you very much for your valuable comments and we have carefully studied the reference you provided. We added the TSA in the manuscript to determine if each outcome needs more studies. The results of the TSA were listed in the Trial sequential analysis section of Results and the Supplementary Figure 5 [Line 387].

Note from Editor- While a trial sequential analysis would be a useful addition this is not a mandatory revision under our formatting guidelines.

Reviewer: 2

Dr. Sameer Bhat, The University of Auckland

Comments to the Author:

Fan and colleagues performed a systematic review and meta-analysis to assess the efficacy and safety of invasive acupuncture for treating postsurgical gastroparesis syndrome after thoracic or abdominal surgery. Overall, the study is well researched and written, with clear objectives and a well-constructed analysis. However, I have a few comments / concerns:

Comment 1. The authors have defined "total effective rate" and "recovery rate" as their primary study outcomes. It is unclear what these relate to, and whether the definitions are homogeneous across the 12 and 8 trials, respectively, which assess these outcomes. Please comment and discuss.

Response 1. Thank you. We have explained this in detail in our manuscript. At present, we have added this part in the Outcome assessment of Methods section [Line 155]. In the trials included in this meta-analysis, the evaluation criteria for the total effective rate (TER) and the recovery rate (RCR) of postsurgical gastroparesis syndrome (PGS) were centered on these proposed by the China Association of Chinese Medicine (CACM), which are the combination of objective indicators (gastric

juice volume, gastrointestinal radiography) and subjective indicators (whether nausea, vomiting, and bloating disappear).

Recovered: Objective indicators: There was no gastric juice volume, and gastrointestinal radiography indicated good gastric motility. Subjective indicators: Symptoms such as nausea, vomiting, and bloating disappeared after eating without gastrointestinal decompression.

Effective: Objective indicators: The gastric juice volume decreased significantly ($< 150\text{ml/d}$), and gastrointestinal radiography indicated gastric hypomotility slightly. Subjective indicators: Symptoms such as nausea, vomiting, and bloating appeared without gastrointestinal decompression, and inability to eat.

TER refers to the proportion of recovered and effective people in all participants. RCR refers to the proportion of recovered people in all participants.

Similar situations have been found in published meta-analysis literatures in English [1, 2].

Comment 2. The authors have concluded that acupuncture may be safe to perform due to improvements in clinical symptoms and lack of demonstrated serious adverse events. I would suggest that a more cautious and conservative statement is used (or this is removed from the manuscript conclusion entirely) given that adverse outcomes associated with acupuncture were reported in only two trials.

Response 2. We strongly agree with you that there were too few literatures on "adverse events related to acupuncture" to reach the conclusion that "acupuncture may be safe", and more trials are needed to study it. Therefore, we removed this conclusion from the Abstract section and the Discussion section. And we described it more cautious and mentioned it as a limitation of our study in the Discussion section [Line 419, 486].

Comment 3. Furthermore, heterogeneity among trials which used the clinical symptom score was very large ($I^2=100\%$), despite the authors' subgroup analysis based on the site of surgery.

Response 3. We initially considered that the large heterogeneity of trials reported the clinical symptom score was mainly due to the inconsistent scoring criteria used in them. We attempted a meta-analysis of the three trials which adopted completely consistent scoring criteria. Unfortunately, there was still significant heterogeneity. This may be because the CSS was subjective indicator and there were possibilities of bias. At present, there are too few relevant trials to analyze the reasons of heterogeneity. This requires more high-quality trails for further study and we have discussed it in the Discussion section and mentioned it as a limitation in the meta-analysis [Line 408, 479].

Thank you again for raising important questions to help us to improve the quality of the manuscript. We have tried our best to revise and explain these comments / concerns. If there are any more, we would be very happy to revise and explain them.

Reviewer: 3

Mr. Hisashi Noma, The Institute of Statistical Mathematics

Comments to the Author:

I am a statistical reviewer. There are several relevant concerns for the statistical methods.

Comment 1. The authors used "RR" as abbreviations of "risk ratio" and "recovery rate". It is confusable.

Response 1. Thank you very much for pointing out the issue that we have neglected. To avoid confusion, we used "RCR", not "RR", as the abbreviation of "Recover Rate" [Line 155], and the manuscript has been revised.

Comment 2. The authors determined the uses of FE or RE models by whether I^2 exceeds 50% or not. However, I^2 statistic is solely a summary statistic and it involves uncertainty. It is not an adequate reason. Please re-check the Cochrane Handbook for this issue.

Response 2. After re-checking the Cochrane Handbook and researching literatures [3], we added the Chi-squared test in the Methods section [Line 189] and added the test results in the Abstract and

Outcomes sections. And here we did not list the changes but marked in red-colored text in the revised manuscript. After the revision, we found that there were no changes in determining the uses of FE or RE models.

Comment 3. Line 145: They noted “Subgroup studies were conducted”, but I guess it would be “subgroup analyses”.

Response 3. Thank you very much for your reminding. We are very ashamed of the cursoriness. It has been revised [Line 191].

Comment 4. In Statistical Analyses section, they should mention they used the Egger test for the publication bias test.

Response 4. We have mentioned that the Egger test was used for the publication bias test in the Statistical analysis section [Line 195].

Comment 5. Although the Egger test was not significant, the funnel plot in S-Figure 4 was substantially asymmetric. It might indicate publication biases, and it should be at least mentioned as a limitation of this study in Discussion section.

Response 5. Indeed, as you mentioned, the funnel plot was asymmetric, indicating that it might have publication bias. We have described it more rigorously in the Publication bias of Results section [Line 384] and mentioned it as a limitation of this study in Discussion section [Line 481].

Comment 6. The English should be checked by native English speakers.

Response 6. We apologize for the poor language of our manuscript. We worked on the manuscript for a long time and repeated addition and removal of sentences obviously led to poor readability. We have invited native English speakers for language corrections to make our manuscript readable. These revises will not influence the content and framework of the manuscript. And here we did not list the changes but marked in red-colored text in the revised manuscript. We really hope that the language level has been improved substantially.

References

- [1] Fu Q, Yang H, Zhang L, et al. Traditional Chinese medicine foot bath combined with acupoint massage for the treatment of diabetic peripheral neuropathy: A systematic review and meta-analysis of 31 RCTs. *Diabetes Metab Res Rev.* 2020;36:e3218.
- [2] Wang T, Xu C, Pan K, Xiong H. Acupuncture and moxibustion for chronic fatigue syndrome in traditional Chinese medicine: a systematic review and meta-analysis. *BMC Complement Altern Med.* 2017;17:163.
- [3] Wu Q, Yuan L, Qiu H, et al. Efficacy and safety of omalizumab in chronic rhinosinusitis with nasal polyps: a systematic review and meta-analysis of randomised controlled trials. *BMJ Open.* 2021;11:e047344.

VERSION 2 – REVIEW

REVIEWER	He, Jing Guangzhou Women and Children's Medical Center, Department of Pediatric Surgery
REVIEW RETURNED	18-May-2023

GENERAL COMMENTS	Acceptable.
-------------

REVIEWER	Noma, Hisashi The Institute of Statistical Mathematics, Department of Data Science
-----------------	---

REVIEW RETURNED	16-May-2023
GENERAL COMMENTS	Thank you for considering my comments. All of my concerns were adequately addressed.

VERSION 2 – AUTHOR RESPONSE